# Effects of the Use of Rice Grain on Growth Performances, Blood Metabolites, Rumen Fermentation, and Rumen Microbial Community in Fattening Hanwoo Steers

**DOI:** 10.3390/ani13182988

**Published:** 2023-09-21

**Authors:** Daekyum Yoo, Sungjae Yang, Hanbeen Kim, Joonbeom Moon, Jakyeom Seo

**Affiliations:** Department of Animal Science, Life and Industry Convergence Research Institute, Pusan National University, Miryang 50463, Republic of Korea; fbrjsgud@pusan.ac.kr (D.Y.); sj@pusan.ac.kr (S.Y.); khb3850@pusan.ac.kr (H.K.); mantis0044@pusan.ac.kr (J.M.)

**Keywords:** corn grain, growth performance, Hanwoo steer, rice grain, total mixed ration

## Abstract

**Simple Summary:**

As the price of imported corn rises due to environmental issues and the growing demand for bio-ethanol production, animal nutritionists are actively seeking alternative energy sources to corn. While previous research has shown that rice grain provides a comparable nutritional value to corn, it is noted that most of the studies on rice grain as a starch source for ruminants have been conducted on dairy cattle. The objective of this study is to assess the effects of total mixed ration (TMR) containing rice grains on the growth performance, blood metabolites, rumen fermentation, and rumen microbial community in fattening Hanwoo steers. Our results demonstrated that rice grain had no adverse effects on the growth performance of Hanwoo steers, rumen fermentation characteristics, and blood metabolites. In conclusion, this study suggests that utilizing rice grains as the primary starch source in TMR may serve as an alternative option for fattening Hanwoo steers.

**Abstract:**

This study aimed to assess the influence of rice grain in the total mixed ration (TMR) on the growth performance, blood metabolites, rumen fermentation, and rumen microbial community of fattening Hanwoo steers. Two experimental diets were prepared: (i) a TMR containing 33% dry matter (DM) corn grains (Corn TMR) and (ii) a TMR containing 33% DM rice grains (Rice TMR). Twenty-two Hanwoo steers (body weight [BW], 498 ± 32 kg; months, 17 ± 0.5) were distributed into two treatment groups in a completely randomized block design according to BW. The Rice TMR group had a higher final BW and DM intake (DMI) compared to those in the Corn TMR group (*p* < 0.01). However, no difference was observed in the average daily gain (ADG) and feed conversion ratio (FCR) between the two treatments. For the rumen fermentation parameters, the molar portion of butyrate in the Rice TMR was higher than in the Corn TMR (*p* < 0.01). *Streptococcus bovis* tended to be higher in the Rice TMR (*p* = 0.09). The results of this study suggest that using rice grain as the primary starch source in TMRs may be an alternative option for fattening Hanwoo steers.

## 1. Introduction

In South Korea, the ruminant livestock population primarily consists of Hanwoo cattle, which account for over 75% of the total, while sheep and goats combined constitute approximately 12% (https://amis.rda.go.kr, 14 September 2023). Consequently, we chose to use domestically raised Hanwoo cattle, the most extensively bred beef cattle in South Korea, as the subjects of our experiment. Increasing the high-starch content in a diet during the finishing period has been known to be an important feeding strategy to produce high-marbled beef because finishing beef cattle require higher energy from non-fiber carbohydrates (NFC) to accumulate intra-muscular fat in their muscles via lipogenesis [1,2]. Corn is a well-known source of NFC and has traditionally been widely used in the animal industry. Recently, the cost of imported corn has sharply increased due to environmental factors such as war and climate change, as well as an increased demand for livestock farming. Furthermore, the growing demand for bio-ethanol production as a replacement for fossil fuels to reduce greenhouse gas emissions has also contributed to the rising cost of corn. South Korea ranks 110th in global corn production and heavily relies on imports (https://www.atlasbig.com, 5 September 2023). Consequently, Korean animal nutritionists are researching suitable energy sources to replace corn. South Korea ranks 16th in global rice grain production (https://www.atlasbig.com, 5 September 2023). Rice grain is a valuable starch source with a nutritional value comparable to that of corn [3]. However, the utilization of rice grains as an animal feed source has been limited in the South Korean animal industry due to their higher cost compared to corn, as well as the traditional perception that rice is primarily intended for human consumption rather than for animal feed. Meanwhile, the South Korean government has initiated projects to utilize excess stockpiled rice as a feed ingredient for livestock due to the surplus of rice in storage since 2017. The reason is that stored rice has a very low level of human palatability, and significant storage costs arise for the stored rice.

In our previous study [3], we used a total mixed ration (TMR) containing 20% rice grain instead of corn and reported that the TMR containing rice grain did not have any negative effects on growth performances in growing Hanwoo steers *in vivo* and rumen fermentation characteristics *in vitro*. A further study focusing on the effects of rice grain feeding on fattening beef would be necessary to fully understand the potential of rice as a feed ingredient for all stages of beef production. Furthermore, to the best of our knowledge, most ruminant research involving rice grain as a starch source has been conducted on dairy cattle [4,5]. We hypothesized that rice grain could be utilized as an alternative source of starch feed during the fattening period for Hanwoo steers. Therefore, an *in vivo* trial was conducted to investigate the effects of rice grain usage on growth performance, blood metabolites, and rumen parameters in fattening Hanwoo steers. In the present study, the additional amount of rice grain was increased compared to the growing beef feeding trial because the previous study used over 30% of rice grain instead of corn in dairy cattle [4,5].

## 2. Materials and Methods

The protocols for animal use in this study were reviewed and approved by the Animal Research Ethics Committee of Pusan University (PNU-2020-2827).

### 2.1. Formulation of Experimental Diets and Chemical Analysis

The primary components of the two experimental TMRs included a commercial concentrate mix (Nonghyup Feed Co., Miryang, Republic of Korea), alfalfa, corn flakes, timothy, and rice. Table 1 presents the formulation and chemical composition of the experimental diets. Both diets were formulated to achieve or surpass the recommended target average daily gain (ADG) of 0.8 kg/day as specified by NASEM [6]. Prior to chemical analysis, we ground all feed and experimental diets using a cyclone mill (Foss Tecator Cyclotec 1093, Foss, Hillerød, Denmark) equipped with a 1 mm screen and subsequently dried them at 65 °C for 72 h. Dry matter (DM) was determined using the method described by the National Forage Testing Association [7]. The crude protein (CP, #990.03), acid detergent fiber (ADF, #973.18), ether extract (EE, #920.39), and ash (#942.05) were determined following the methods outlined in AOAC [8]. Lignin and neutral detergent fiber (aNDF) were determined following the method described by Van Soest et al. [9]. The aNDF was determined using heat-stable α-amylase and included residual ash content. The net energy required for maintenance and total digestible nutrients of the experimental diet were calculated using the NASEM [6].

### 2.2. In Vivo Experimental Design

Twenty-two Hanwoo steers (body weight [BW] of 498 ± 32 kg and 17 ± 0.5 months old, Pusan National University Farm) were randomly assigned to two treatments (Corn TMR vs. Rice TMR) using a completely randomized block design based on BW. Each group had three pens (5 m × 10 m), and cattle of similar BW were placed in groups of four, four, and three in each pen, respectively. An automated water machine was installed in each pen. Both diets were produced weekly and served to the steers twice daily at 08:00 and 17:00 h, allowing for ad libitum consumption with a target of 10% feed refusal. The experiment lasted for a total of twelve weeks. We calculated the daily DM intake (DMI) by measuring the daily amount of feed provided and the amount refused in each pen. During the *in vivo* trial, all steers were weighed on the first day and at four-week intervals to measure their ADG and feed conversion rate (FCR).

### 2.3. Rumen Fermentation Characteristics

During the experimental period, rumen fluid was collected from all steers using an oral stomach tube before morning feeding every three weeks for two days and instantly transferred to the laboratory (Figure 1). The ruminal pH was determined using a pH meter (FP20, Mettler Toledo, Columbus, OH, USA). Subsequently, the rumen fluid was centrifuged at 3500× rpm for 20 min at 4 °C. For the volatile fatty acids (VFA) assay, 1 mL of the supernatant was acidified using 200 µL of 25% metaphosphoric acid, while for ammonia nitrogen (NH_3_-N) analysis, 1 mL of the supernatant was acidified with 200 µL of 0.2 M sulfuric acid. Both supernatants were kept at −80 °C until they were analyzed for VFA and NH_3_-N. The sample intended for VFA analysis was thawed at 4 °C and subjected to centrifugation at 20,000× *g* for 15 min. Subsequently, 200 μL of the resulting supernatant was diluted with 800 μL of 99% anhydrous ethyl alcohol. The concentration of VFA was determined using a gas chromatograph (Agilent 7890A, Agilent Technology, Santa Clara, CA, USA) following the method described in the study by Yoo et al. [10]. The NH_3_-N concentration was determined with certain modifications to the method proposed by Chaney and Marbach [11], and the assessment was conducted following the procedure outlined by Yoo et al. [10].

### 2.4. Analysis of Blood Metabolites 

During the experimental period, blood was drawn before feeding in the morning every three weeks (Figure 1). Blood samples were collected from the jugular vein of each steer and placed in serum tubes (BD Vacutainer; BD and Co., Franklin Lakes, NJ, USA) containing coagulation activators. After centrifuging the blood in serum tubes at 2500× *g* for 15 min at 4 °C, various serum parameters including creatinine (Crea), inorganic phosphate (IP), alanine aminotransferase (ALT), albumin (Alb), blood urea nitrogen (BUN), triglyceride (TG), calcium (Ca), aspartate aminotransferase (AST), magnesium (Mg), total cholesterol (T-Chol), glucose (Glu), and total protein (TP) were determined using the method described by Yoo et al. [12].

### 2.5. Partial Least Square–Discriminant Analysis (PLS–DA) 

PLS–DA analysis was carried out using Metaboanalyst (version 5.0; http://www.metaboanalyst.ca 15 March 2023). PLS–DA was applied to separate metabolite data between groups. Before conducting PLS–DA, the results of rumen fermentation and blood metabolites underwent sample normalization, data transformation, and data scaling using the log and pareto functions [13].

### 2.6. Total DNA Extraction and Real-Time Quantitative PCR 

Total DNA was extracted from the stored pellet at −80 °C using the repeated bead beating plus column (RBB+C) method [14]. Subsequently, the concentration and purity of the total DNA were evaluated using NanoDrop (ND-1000, Thermo Fisher, Waltham, MA, USA).

Real-time quantitative PCR analysis was conducted using the CFX 96 Touch system (Bio-Rad Laboratories, Inc., Hercules, CA, USA). Primer sequence information of rumen microorganisms is presented in Table 2. The experiment was conducted in triplicate with a reaction volume of 20 µL and sealed with an optical adhesive film on an optical reaction plate. Each reaction mixture included the following components: 13.3 μL of PCR-grade water, 1 μL of genomic DNA (diluted 10-fold), 1 μL of Evagreen (SolGent Co., Ltd., Daejeon, Republic of Korea), 2 μL of 10× buffer (BioFACT, Daejeon, Republic of Korea), 1 μL of forward primer (10 μM), 1 μL of reverse primer (10 μM), 0.5 μL of 10 mM dNTP Mix (BioFACT, Daejeon, Republic of Korea), and 0.2 μL of Taq polymerase (BioFACT, Republic of Korea). We conducted real-time PCR following the manufacturer’s instructions as outlined below: initialization for one cycle (95 °C, 10 min); denaturation for 40 cycles (95 °C, 30 s); annealing (60 °C, 30 s); elongation (72 °C, 30 s); and final elongation (72 °C, 5 min). Fluorescence was recorded at each end of the denaturation and extension phases. The specificity of the amplicons was verified by analyzing the dissociation curves of the PCR final products, which were subjected to a temperature gradient from 60 °C to 95 °C in increments of 1 °C every 30 s. To determine the absolute abundance of each microorganism, we obtained a standard plasmid containing the respective target gene sequence through PCR cloning, using the primer sets listed in Table 2. The copy number of each standard primer was calculated [15] and serially diluted by 10-fold. Standard curves and microbial quantities were compared using CFX Manager software (Version 3.1, Bio-Rad Laboratories, Inc., Hercules, CA, USA).

### 2.7. Statistical Analysis

All data were examined for normal distribution using the Shapiro–Wilk test in the SAS 9.4 (SAS Institute Inc., Cary, NC, USA) package. If the data were not normally distributed, a normal distribution was obtained by transforming the data (log, square root, etc.). *In vivo* experimental data were analyzed using the PROC GLIMMIX procedure in SAS 9.4 according to the following model:*Y_ij_* = *μ* + *R_i_* + *T_j_* + *E_ij_*
in which *Y_ij_* represents the response variable, *μ* is the overall mean, *R_i_* is the random effect of the pen within block (*i* = 1 to 3), *T_j_* is fixed effect of treatment (*j* = 1 to 2), and *E_ij_* represents the residual error. When the overall treatment effect was significant, we compared the differences between treatments using Tukey’s test. Statistically significant differences were declared for *p* < 0.05, and a trend was estimated for 0.05 ≤ *p* < 0.10.

## 3. Results

### 3.1. Growth Performance

The two feeding groups exhibited numerical differences in initial BW, but no statistically significant difference was observed (*p* = 0.13). The Corn TMR group showed a weight change of 70.7 kg, while the Rice TMR group exhibited a weight change of 77.9 kg. The final BW and DMI were higher in the Rice TMR-fed group compared to the Corn TMR-fed group (both *p* < 0.01, Table 3). No differences (*p* > 0.05) for ADG and FCR occurred between experimental groups. 

### 3.2. Rumen Fermentation Characteristics

For the rumen fermentation characteristics, the NH_3_-N, total VFA, and pH were not affected by the treatments (Table 4). The butyrate proportion of VFA was higher in the Rice TMR group (*p* < 0.01, Table 4). No differences were observed in the proportions of acetate and propionate, as well as in the acetate-to-propionate (A:P) ratio between the two groups. 

### 3.3. Blood Metabolites

The Rice TMR led to an increase in the blood levels of AST and BUN compared to the Corn TMR (*p* < 0.05 and *p* < 0.01, Table 5). The concentration of serum TP tended to be higher in the Rice TMR-fed group compared to the Corn TMR-fed group (*p* = 0.07, Table 5). No differences were observed in ALT, Ca, Glu, Alb, IP, Mg, Crea, T-Chol, and TG between the treatments. 

### 3.4. PLS−DA

The results of the PLS−DA for rumen fermentation and blood metabolites are presented in Figure 2. Metabolites in rumen fermentation (component 1, 36.4%; component 2, 29.1%) and blood (component 1, 29.3%; component 2, 28.9%) did not exhibit a clear distinction between the two dietary groups.

### 3.5. Microorganism

In the microbial counts, the absolute abundances of general bacteria, fungi, and ciliate protozoa were expressed in log copies per milliliter of rumen fluid. The numbers of general bacteria, fungi, and ciliate protozoa did not differ between treatments (Table 6). In the bacterial composition, the relative abundances of *P. ruminicola*, *R. albus*, *F. succinogenes*, *R. flavefaciens,* and *S. ruminantium* were similar between Corn and Rice TMRs. *S. bovis* showed a higher tendency in the Rice TMR group compared to the Corn TMR group (*p* = 0.09, Table 6), while *B. fibrisolvens* was higher in the Corn TMR group than in the Rice TMR group (*p* < 0.01, Table 6).

## 4. Discussion

We hypothesized that rice grain could be utilized as an alternative source of starch feed during the fattening period for Hanwoo steers. The selection of the rice grain level in our study, set at 33% DM, was a carefully considered decision aimed at optimizing animal performance while minimizing potential adverse effects on rumen fermentation in fattening beef cattle. In the context of dairy cattle, previous research has shown that incorporating brown rice grain at levels of 31% and 33% in the TMR did not yield adverse effects on DMI, milk yield, and milk composition when compared to the control TMR [4,21]. However, when the diet included rice grain at a 40% level, it led to a reduction in both DMI and milk yield in dairy cattle [5]. White et al. [22] reported a decrease in both DMI and ADG when a diet for beef cattle included 60% rice grain. In our study, the two feeding groups exhibited numerical differences in initial BW, but no statistically significant difference was observed. DMI was significantly higher in the Rice TMR group compared to the Corn TMR group. Although the ADG showed a numerical increase in the Rice TMR group, no significant difference was observed. Based on these findings, the difference in final live weight is approximately 24 kg, which is higher than the initial weight difference of 17 kg. This suggests that the inclusion of 33% DM rice grain in the TMR for Hanwoo beef cattle might have positively influenced the growth performance, with these effects more likely attributed to the rice grain inclusion rather than differences in the initial BW.

Yoo et al. [10] observed no difference in CP digestibility between corn TMR and rice TMR, and they also found no significant difference in NH_3_-N concentration, which was positively correlated with CP digestibility. Similar to the findings of Yoo et al. [10], our study showed no significant difference in NH_3_-N concentration between the groups fed with Corn TMR (7.12 mg/dL) and Rice TMR (7.33 mg/dL). Moreover, the ruminal NH_3_-N concentration in our study fell within the previously suggested optimal range for proper rumen fermentation (3.3 to 8.5 mg/dL) [23]. Regarding the VFA composition, the proportion of butyrate was higher in the Rice TMR group than in the Corn TMR group. This differed with the findings of Yoo et al. [10], who reported no difference in butyrate proportion between the rice TMR and corn TMR. Ha et al. [24] reported that the normal range for butyrate is 80–150 mmol/mol; therefore, the butyrate proportion in the Rice TMR (149 mmol/mol) fell within the normal range. The total VFA production in this experiment was lower than that reported in previous studies [4,5]. This discrepancy in the total VFA production might be attributed to the difference in rumen fluid collection time because we collected rumen fluid before the morning feeding when the total VFA concentration is typically lowest during the day, owing to unavoidable experimental environmental factors. 

Blood metabolites are substances produced during the metabolic process that can be used as indicators of an animal’s health status, nutrient utilization, and overall metabolism [25]. The concentrations of IP and Ca, mainly controlled by parathyroid hormones, are associated with bone growth [25]. Kwon et al. [25] reported the normal ranges for IP (6.0–8.2 mg/dL) and Ca (9.9–10.3 mg/dL), which are similar to our results. Blood cholesterol concentration may be associated with intramuscular fat deposition, and an increase in blood cholesterol can enhance the marbling score in beef meat [26]. In our study, no significant difference existed in blood cholesterol levels between the two groups. This suggests that substituting corn with rice grains in the diet during the fattening phase of Hanwoo steers may not negatively impact marbling formation. The BUN concentration in the Rice TMR group was higher than that in the Corn TMR group. This result might be attributed to differences in the DMI between the treatment groups. In this study, steers fed the Rice TMR had higher BUN levels as the DMI increased, which is consistent with the findings of Huntington et al. [27], who reported that an increased intake of concentrate feed can lead to higher BUN in fattening steers. The serum concentrations of AST and ALT serve as reliable indicators for hepatic diagnostics in animals [28]. The AST in the Rice TMR was higher than that in the Corn TMR (Corn TMR, 70.7 U/L; Rice TMR, 79.5 U/L), but there was no significant difference in ALT between the two groups. Previous studies had suggested ranges for AST (60.6–88.7 U/L) and ALT (18.9–24.1 U/L) levels in the fattening phase of Hanwoo steers [29,30,31]. Since both ALT and AST levels were within the ranges indicated by previous research in this study, it is presumed that substituting rice grain for corn had no adverse effect on liver function. The PLS–DA based on serum metabolites did not show a clear separation between the two groups, indicating that rice grains could potentially be used as a substitute for corn in the diet (Figure 2b).

In this study, we investigated the absolute abundances of general bacteria, ciliate protozoa, and fungi in the rumen. Additionally, we measured the relative abundance of the main rumen carbohydrate-degrading bacteria. *S. bovis*, a main starch-utilizing bacterium among bacteria that degrade nonstructural carbohydrates in the rumen, which showed a high tendency in the Rice TMR. Cotta [32] reported that *S. bovis* grows very rapidly in starch-rich environments. Therefore, the trend toward higher *S. bovis* in the Rice TMR group might be linked with the higher DMI in the steers fed the Rice TMR. Among the structural carbohydrate-degrading bacteria in the rumen, we found no differences between the two groups in the relative abundance of major fibrinolytic bacteria, including *R. albus*, *F. succinogenes*, and *R. flavefaciens* [33]. Scheilbler et al. [21] observed that the 33% brown rice grain level of TMR showed similar aNDF digestibility to the control TMR. Therefore, it is estimated that the level of 33% DM of rice grain in TMR did not have a negative effect on fiber digestibility. The relative abundance of *B. fibrisolvens* was higher in the Corn TMR but the butyrate proportion in the rumen fluid was lower. Although the major metabolic end product of *B. fibrisolvens* is butyrate [34], the difference in butyrate proportion might not be explained by only *B. fibrisolvens* abundance because the rumen is a unique habitat consisting of such diverse microbes. Overall, there is a limit to explaining nutritional and physiological parts only by analyzing some microorganisms, and it is judged that a microbiome analysis using metagenomic techniques is necessary.

## 5. Conclusions

In conclusion, our study supports the use of rice grain at a 33% DM level in TMR as an alternative starch feed for fattening Hanwoo cattle. Overall, feeding with rice grain did not exhibit any negative effects on the growth performance, rumen fermentation, and blood metabolites of fattening Hanwoo steers. A microbiome analysis using metagenomic techniques is necessary for a more in-depth understanding of the nutritional and physiological aspects.

## Figures and Tables

**Figure 1 animals-13-02988-f001:**
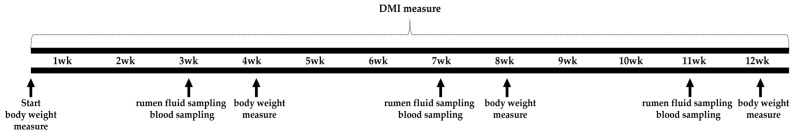
Brief schedule of *in vivo* experiment.

**Figure 2 animals-13-02988-f002:**
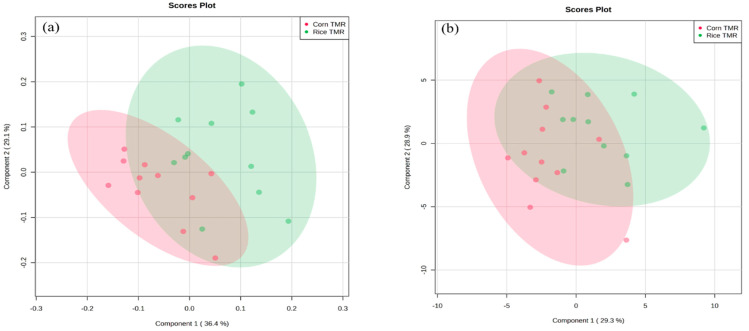
PLS-DA scores for (**a**) rumen fermentation and (**b**) blood metabolites in fattening steers fed with Corn TMR and Rice TMR. On the score plot, each point represents a sample, with red dots representing Corn TMR (*n* = 11) and green dots representing Rice TMR (*n* = 11). The shaded ellipses on the PLS-DA score plot represent the 95% confidence interval estimated from the scores. The *x*-axis and *y*-axis represent the variance associated with components 1 and 2, respectively.

**Table 1 animals-13-02988-t001:** Experimental diet formulation and chemical composition.

Items	Corn TMR	Rice TMR
Ingredients (% DM)		
Commercial concentrate mix ^1^	36.0	36.0
Corn flake	33.0	0
Rice grain	0	33.0
Timothy	23.5	23.5
Alfalfa	7.0	730
Vitamin and mineral mix ^2^	0.5	0.5
Chemical composition ^3^		
DM (% as fed)	65.0	65.0
CP (% DM)	14.9	14.8
aNDF (% DM)	30.6	29.5
ADF (% DM)	19.5	18.5
Lignin (% DM)	4.45	4.17
EE (% DM)	3.50	3.19
Ash (% DM)	5.22	5.42
TDN (% DM)	72.4	73.8
NEm (Mcal/kg of DM)	1.67	1.71

^1^ Commercial concentration mix was purchased from local feed company (Nonghyup Feed, Co. Ltd., Miryang, Republic of Korea). ^2^ 0.02 g/kg of Cu, 90 mg/kg of Mn, 0.1 g/kg of Zn, 0.25 g/kg of Fe, 0.004 g/kg of I, 0.004 g/kg of Se, 20.86 IU/kg of vitamin E, 33,330,000 IU/kg of vitamin A, and 40,000,000 IU/kg of vitamin D. ^3^ CP, crude protein; aNDF, neutral detergent fiber; EE, ether extract; ADF, acid detergent fiber; TDN, total digestible nutrients; and NEm, net energy for maintenance calculated using the equation from NASEM [6].

**Table 2 animals-13-02988-t002:** Polymerase chain reaction primers used in this study.

Target Species	Primer	Sequence (5′ → 3′)	Size (bp)	Efficiency ^1^	References
General bacteria	F	CGGCAACGAGCGCAACCC	130	1.992	[16]
	R	CCATTGTAGCACGTGTGTAGCC			
Ciliate protozoa	F	GCTTTCGWTGGTAGTGTATT	223	1.947	[17]
	R	CTTGCCCTCYAATCGTWCT			
Fungi	F	GAGGAAGTAAAAGTCGTAACAAGGTTTC	120	2.079	[16]
	R	CAAATTCACAAAGGGTAGGATGATT			
*Fibrobacter succinogenes*	F	CCCTAAAAGCAGTCTTAGTTCG	121	1.925	[16]
	R	CCTCCTTGCGGTTAGAACA			
*Ruminococcus albus*	F	CCCTAAAAGCAGTCTTAGTTCG	176	2.067	[18]
	R	CCTCCTTGCGGTTAGAACA			
*Ruminococcus flavefaciens*	F	CGAACGGAGATAATTTGAGTTTACTTAGG	132	2.029	[16]
	R	CGGTCTCTGTATGTTATGAGGTATTACC			
*Butyrivibrio fibrisolvens*	F	ACCGCATAAGCGCACGGA	65	2.006	[19]
	R	CGGGTCCATCTTGTACCGATAAAT			
*Streptococcus bovis*	F	TTCCTAGAGATAGGAAGTTTCTTCGG	127	2.005	[20]
	R	ATGATGGCAACTAACAATAGGGGT			
*Selenomonas ruminantium*	F	GGCGGGAAGGCAAGTCAGTC	83	2.188	[20]
	R	CCTCTCCTGCACTCAAGAAAGACAG			
*Prevotella ruminicola*	F	GCGAAAGTCGGATTAATGCTCTATG	78	1.946	[20]
	R	CCCATCCTATAGCGGTAAACCTTTG			

^1^ Efficiency is calculated as [10^−1^/slope]; bp, base pair.

**Table 3 animals-13-02988-t003:** Effect of the use of rice grain in total mixed ration (TMR) on growth performance in fattening steers.

Items	Corn TMR	Rice TMR	SEM	*p*-Value
Initial BW (kg)	488	506	5.8	0.13
Final BW (kg)	559	583	8.0	<0.01
ADG (g/d)	786	867	62.2	0.21
DMI (kg/d)	8.00	8.99	0.059	<0.01
FCR	10.6	10.7	0.73	0.99

ADG, average daily gain; BW, body weight; DMI, dry matter intake; FCR, feed conversion ratio; and SEM, standard error of the mean.

**Table 4 animals-13-02988-t004:** Effect of the use of rice grain in total mixed ration (TMR) on rumen fermentation characteristics in fattening steers.

Items	Corn TMR	Rice TMR	SEM	*p*-Value
TVFA (mM)	54	59	5.5	0.32
Acetate (mmol/mol)	633	622	15.9	0.50
Propionate (mmol/mol)	181	168	8.0	0.11
Butyrate (mmol/mol)	121	149	6.4	<0.01
A:P ratio	3.55	3.88	0.201	0.11
NH_3_-N (mg/dL)	7.12	7.33	0.668	0.76
pH	7.13	7.01	0.100	0.22

A:P ratio, acetate-to-propionate ratio; NH_3_-N, ammonia nitrogen; TVFA, total volatile fatty acids; and SEM, standard error of the mean.

**Table 5 animals-13-02988-t005:** Effect of the use of rice grain in total mixed ration (TMR) on blood metabolites in fattening steers.

Items	Corn TMR	Rice TMR	SEM	*p*-Value
ALT (U/L)	22.7	22.3	0.83	0.66
Alb (g/dL)	3.56	3.57	0.028	0.72
AST (U/L)	70.7	79.5	3.86	<0.05
BUN (mg/dL)	15.5	17.9	0.43	<0.01
Ca (mg/dL)	10.3	10.3	0.08	0.29
Crea (mg/dL)	1.35	1.32	0.003	0.39
Glu (mg/dL)	72.1	75.8	2.70	0.18
IP (mg/dL)	7.45	7.32	0.186	0.50
Mg (mg/dL)	2.33	2.28	0.041	0.19
T-Chol (mg/dL)	130.4	118.7	5.95	0.54
TG (mg/dL)	18.6	18.9	1.46	0.82
TP (g/dL)	6.94	7.10	0.084	0.07

ALT, alanine aminotransferase; Glu, glucose; Alb, albumin; AST, aspartate aminotransferase; IP, inorganic phosphorus; Ca, calcium; TG, triglyceride; Crea, creatinine; BUN, blood urea nitrogen; Mg, magnesium; T-Chol, total cholesterol; TP, total protein; and SEM, standard error of the mean.

**Table 6 animals-13-02988-t006:** Effect of the use of rice grain in total mixed ration (TMR) on the microbial abundance in fattening steers.

Items	Corn TMR	Rice TMR	SEM	*p*-Value
Absolute abundance ^1^				
General bacteria	6.24	6.38	0.910	0.88
Ciliate protozoa	3.35	3.74	0.958	0.69
Fungi	5.40	5.96	1.059	0.60
Relative abundance (% General bacteria)				
*Ruminococcus albus*	0.35	0.33	0.044	0.76
*Fibrobacter succinogenes*	3.14	3.75	0.575	0.29
*Ruminococcus flavefaciens*	0.14	0.10	0.032	0.17
*Butyrivibrio fibrisolvens*	0.065	0.033	0.0074	<0.01
*Streptococcus bovis*	0.018	0.022	0.0024	0.09
*Selenomonas ruminantium*	2.35	2.09	0.260	0.34
*Prevotella rumnicola*	6.75	7.58	0.707	0.25

^1^ Fungi, ×10^7^ copies/mL of rumen fluid; ciliate protozoa, ×10^9^ copies/mL of rumen fluid; general bacteria, ×10^10^ copies/mL of rumen fluid; and SEM, standard error of the mean.

## Data Availability

The data presented in this study are available on request from the corresponding author.

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
