# Peer review of "Effects of the Use of Rice Grain on Growth Performances, Blood Metabolites, Rumen Fermentation, and Rumen Microbial Community in Fattening Hanwoo Steers"

_animals, 2023, doi:10.3390/ani13182988_

Round 1
Reviewer 1 Report
This paper is well-written, results are clear, and discussion is scientifically sound.
This paper could be published after revision of following points:
Tables 3,4,5; please try to use effective digits;
- E.g., Initial BW 488.3 to 488; p-value 0.1306 to 0.13.
Author Response
Thank you for your review and comment. We highlighted all changed things using yellow color in the manuscript

Reviewer 2 Report
The manuscript entitled "Effects of the Use of Rice Grain on Growth Performances, Blood Metabolites, Rumen Fermentation, and Rumen Microbial Community in Fattening Hanwoo Steers" by Dae Kyum Yoo et al. to evaluate the effects of the use of rice grain and corn grains on productivity performance, blood metabolites, rumen fermentation, and rumen microbial community in fattening Hanwoo steers. The study has some problem about the initial BW (kg) difference. The initial BW (kg) is 488.3 for Corn TMR, and is 505.7 for Rice TMR. The difference is 17.4 kg for the initial BW (kg) between Corn TMR and Rice TMR groups. In addition, the author should add a normal group about diet formula used in many fattening Hanwoo Steers farm. These important productivity performance data are showed owing to initial BW (kg) difference, nor rice grain. Therefore, this study design has some flaws, and the manuscript should be rejected.
Author Response
Thank you for your review and comment.

Reviewer 3 Report
In the present study, the authors evaluated the use of rice grain in TMR on growth performances, blood metabolites and rumen microbial community in fattening Hanwoo steers. This work adds to the evidence that rice grain can be used as the source of starch in TMR. I recommend this for publication after the authors have addressed the following.
1. Line 25: Add units to body weight.
2. Line 37: Replace “Increasing” with “increasing”.
3. Line 99: Please check the 0800 and 1700 format for errors.
4. Line 204: Please add a brief analysis of Glu, T-Chol and TP in Table 5.
5. Line 287: The sentence is incomplete. Please check it.
6. Table 1: Briefly describe the methods used to detect EE.
7. Table 3: Add units to DMI.
8. Table 4: Please briefly describe the steps used to measure acetate, propionate and butyrate.
Author Response
Thank you for your review and comment. We highlighted all changed things using green color in the manuscript

Round 2
Reviewer 2 Report
In present, I have no suggestions.
Author Response
Thank you for reviewing my manuscript
